# EXtra-Xwiz: A Tool to Streamline Serial Femtosecond Crystallography Workflows at European XFEL

Oleksii Turkot [1,†], Fabio Dall'Antonia [1,†], Richard J. Bean [1], Juncheng E [1], Hans Fangohr [1,‡], Danilo E. Ferreira de Lima [1], Sravya Kantamneni [1], Henry J. Kirkwood [1,§], Faisal H. M. Koua [1], Adrian P. Mancuso [1,2,∥], Diogo V. M. Melo [1], Adam Round [1], Michael Schuh [1], Egor Sobolev [1], Raphaël de Wijn [1], James J. Wrigley [1] and Luca Gelisio [1,*]

[1] European XFEL, Holzkoppel 4, 22869 Schenefeld, Germany; oleksii.turkot@xfel.eu (O.T.); fabio.dall.antonia@xfel.eu (F.D.); juncheng.e@xfel.eu (J.E.); hans.fangohr@mpsd.mpg.de (H.F.); danilo.enoque.ferreira.de.lima@xfel.eu (D.E.F.d.L.); sravya.kantamneni@xfel.eu (S.K.); faisal.koua@xfel.eu (F.H.M.K.); diogo.melo@xfel.eu (D.V.M.M.); adam.round@xfel.eu (A.R.); michael.schuh@xfel.eu (M.S.); egor.sobolev@xfel.eu (E.S.); raphael.de.wijn@xfel.eu (R.d.W.); james.wrigley@xfel.eu (J.J.W.)
[2] La Trobe Institute for Molecular Science, Bundoora, VIC 3086, Australia
* Correspondence: luca.gelisio@xfel.eu; Tel.: +49-40-8998-6761
† These authors contributed equally to this work.
‡ Current address: Max Planck Institute for the Structure and Dynamics of Matter, 22761 Hamburg, Germany.
§ Current address: PlantTech Research Institute, Tauranga 3110, New Zealand.
∥ Current address: Diamond Light Source, Harwell Science and Innovation Campus, Didcot OX11 0DE, UK.

**Abstract:** X-ray free electron lasers deliver photon pulses that are bright enough to observe diffraction from extremely small crystals at a time scale that outruns their destruction. As crystals are continuously replaced, this technique is termed serial femtosecond crystallography (SFX). Due to its high pulse repetition rate, the European XFEL enables the collection of rich and extensive data sets, which are suited to study various scientific problems, including ultra-fast processes. The enormous data rate, data complexity, and the nature of the pixelized multimodular area detectors at the European XFEL pose severe challenges to users. To streamline the analysis of the SFX data, we developed the semiautomated pipeline *EXtra-Xwiz* around the established *CrystFEL* program suite, thereby processing diffraction patterns on detector frames into structure factors. Here we present *EXtra-Xwiz*, and we introduce its architecture and use by means of a tutorial. Future plans for its development and expansion are also discussed.

**Keywords:** serial femtosecond crystallography; SFX; *EXtra-Xwiz*; pipeline; *CrystFEL*





## 1. Introduction

The structural arrangement of atoms in matter and the nature of their chemical bonds can be deciphered by employing X-ray crystallography. This experimental technique has been pivotal in structural biology (see, e.g., [1–3] and the references therein) and contributes, to date, to approximately 85% of the structures released in the Protein Data Bank (calculated using data from [4,5]). X-ray crystallography requires samples to be present in the form of crystals: these are periodic repetitions of a unique unit cell, which results in Bragg peaks upon exposure to X-rays. These encode part of the information to reconstruct the electron density of the sample. In a simplified description, assuming that the kinematic approximation holds, the overall intensity of the Bragg peaks increases with (i) the squared number of unit cells in the crystal and its degree of perfection, and (ii) the number of photons interacting with the sample (For an extensive description, the reader is referred to, e.g., [6]). The size and quality of the crystals can only be controlled to a limited extent, which is also due to the need to maintain realistic near-physiological conditions when biological

compounds are investigated. The key to elucidate increasingly complex macromolecules, from, e.g., hemoglobin [7] to the ribosome [8], is instrumental to advance structural biology, and it has thus been the exploitation of more and more advanced photon sources. The photon flux generated from modern sources, which are either based on storage rings or X-ray-free electron lasers (XFELs), has enabled the collection of diffraction data that allows up to Ångstrom-level resolution from smaller and smaller crystals, down to the submicrometer size measured by XFELs as a result of their exceptional brilliance. The interaction of crystals with intense X-ray beams results in several physical processes that can lead to permanent radiation damage (see, e.g., [9]). Classical crystallography experiments consist of rotation scans that repeatedly expose the same region of the crystal. Consequently, it accumulates damage during data acquisition. To mitigate this, cryogenic conditions can be employed so as to hamper the processes of radical formation from photoelectrons [10]. Another strategy consists of collecting data from fresh regions of the sample, either of the same crystal or different ones, in a serial fashion (see, e.g., [11]). The latter paradigm has been utilized with XFELs as a single X-ray pulse that typically deposits enough energy to completely destroy the sample [12]. However, the temporal duration of the pulses is short enough—of the order of several femtoseconds—that signals from almost undamaged crystals can be detected, as the timescale of the ion movement is much longer (For a comprehensive introduction to structure determination using XFELs, the reader is referred to [13]). When using XFELs, the crystals are continuously replaced, typically using liquid jets or movable fixed-target stages, and they intercept the X-ray beam with a certain probability (the so-called hit rate). This technique is termed serial femtosecond crystallography (SFX) [14–18]. Furthermore, owing to the femtosecond duration of their pulses, XFELs are exceptional tools for performing time-resolved investigations at resolutions that are not achievable by photon sources based on storage rings [13,19–23].

In parallel to the development of X-ray sources and instrumentation, the computer hardware and software to process data from raw diffraction images to the final structural model have evolved in several aspects, including the development of novel crystallographic methods, the usage of parallel processing computational methods, and the design of graphical interfaces to facilitate and automate the data processing. The processing of X-ray crystallography data commences with the reduction of raw detector frames to a unique set of structure factors [24]. This includes finding Bragg peaks, indexing them, integrating pixel intensities in three dimensions, and averaging the symmetry-equivalent reflection observations with proper scaling. These steps have been implemented in popular software packages such as XDS [25], Mosflm [26], and *DIALS* [27]. Owing to the nature of data collection, serial crystallography requires different algorithms and approaches, and includes a hit-finding step for identifying detector frames containing the signature of diffraction, which is a prerequisite for indexing the reciprocal lattice [17]. Additionally in this case, dedicated software suites have been developed in the last decade. Notably, *DIALS* has been extended to process serial crystallography data [28], and the *CrystFEL* suite [29,30] has been developed. Subsequent data analysis, from structure factors to the final atomic model, requires software for crystallographic phasing, model building from derived electron densities, and its refinement and validation (For a detailed explanation, the reader is referred to, e.g., [31]).

The entire crystallography data processing pipeline consists of the sequential execution of several tasks, which are often performed by different software programs. Given that the input and output data formats, as well as the user experience, might differ greatly across tools, several software pipelines have been developed with the aim of abstracting the complexity and increasing the analysis throughput and automation level [31–34]. In fact, X-ray crystallography beamlines at storage rings are exceptional examples of sophisticated ecosystems, which include state-of-the-art robotics, information systems, and processing tools, thereby allowing for comprehensive automation [31,35,36]. Such simplification empowers inexperienced users to focus on scientific questions. It should be pointed out

that the need for expert knowledge persists in demanding cases, e.g., when the diffraction signal is particularly weak, and extensive parameter optimization is required.

Several challenges are intrinsic to serial crystallography using XFELs, such as the pulse-to-pulse jitter of the X-ray beam in terms of the space, wavelength, and energy, which are typically reflected in the amount of diagnostics necessary to interpret the outcome of the experiment. Additionally, the rate and amount of data collected to solve the scientific problem under investigation, as well as the often-complicated nature of custom-built detectors, further complicate processing and interpretation [15,17,37]. For example, the European XFEL (EuXFEL) [38,39] generates up to 27,000 X-ray pulses per second. A fraction of these is collected by multimodular pixelized area detectors, such as the Adaptive-Gain Integrating Pixel Detector (AGIPD, up to 3520 frames or 14 GiB per second) [40], the Large Pixel Detector (LPD, up to 5100 frames or 10 GiB per second) [41], and the JUNGFRAU detector (up to 160 frames or 1.9 GiB per second) [42], which are synchronized with X-ray pulses. Due to technical reasons, the data acquisition system stores predefined sequences of data from each detector module into separate files in the EuXFEL data format (EXDF), which might be a significant barrier for users, and this approach currently cannot be used directly by popular software like *CrystFEL* (currently at its version 0.10.2). Additionally, the sheer volume of the data to be analyzed makes workflows practically unfeasible unless distributed computing on high-performance computing (HPC) clusters is employed, whose usage is an additional burden to scientists. Finally, photon sources such as the EuXFEL enable the investigation of ultra-fast processes, which are often performed by utilizing some form of excitation of the sample (the so-called pump) to then probe the induced molecular dynamics with the XFEL beam. The cost of this is tedious bookkeeping of the data frame subsets, given the pump–probe patterns and the verification of correct time sampling, and the same applies in general to diagnostic means.

With the aim of abstracting as much complexity as possible so as to allow scientists to focus on their biological question, we developed the *EXtra-Xwiz* [43] pipeline. This allows for a high degree of automation of data analysis workflows through its integration with other services provided at the EuXFEL. In this paper, we introduce *EXtra-Xwiz* and discuss its current status and future goals. In Section 2 we describe the *EXtra-Xwiz* design and architecture, followed by a step-by-step tutorial with an example of processing SFX data in Section 3. Finally, we give an outlook on planned extensions to the pipeline in Section 4.

## 2. Design and Implementation of the EXtra-Xwiz Pipeline

*EXtra-Xwiz* consists of an SFX workflow-managing tool that is bundled with some auxiliary programs. In the simplest scenario, a workflow is run as a linear pipeline that will (i) prepare input data, (ii) distribute the input frames into subsets that can be processed in parallel, (iii) perform merging and scaling of the structure factor intensity observations, and (iv) create crystallographic figures of merit (FOMs) [44].

To accomplish this, *EXtra-Xwiz* mostly utilizes programs from the *CrystFEL* suite. These include the following:

- *indexamajig* — main command-line program for indexing and integrating diffraction patterns;
- *cell_explorer* — a tool for displaying and determining the unit cell parameters of the crystalline sample;
- *partialator* — used for scaling, merging, and postrefining reflections data;
- *check_hkl* — used for calculating FOMs based on the full set of merged reflections, such as completeness, average signal strength, and redundancy;
- *compare_hkl* — used for calculating the FOMs based on the merged reflections split into two sets, such as *R* factors and correlation coefficients.

The full list of the *CrystFEL* programs can be found in the dedicated publication [45] and documentation [46]. *CrystFEL* is freely available at [46].

*EXtra-Xwiz* has been released as an open-source project at [47].

*Structure of the EXtra-Xwiz Pipeline*

A schematic representation of the pipeline is shown in Figure 1. *EXtra-Xwiz* requires a configuration file with parameters for each step, which are organized into sections.

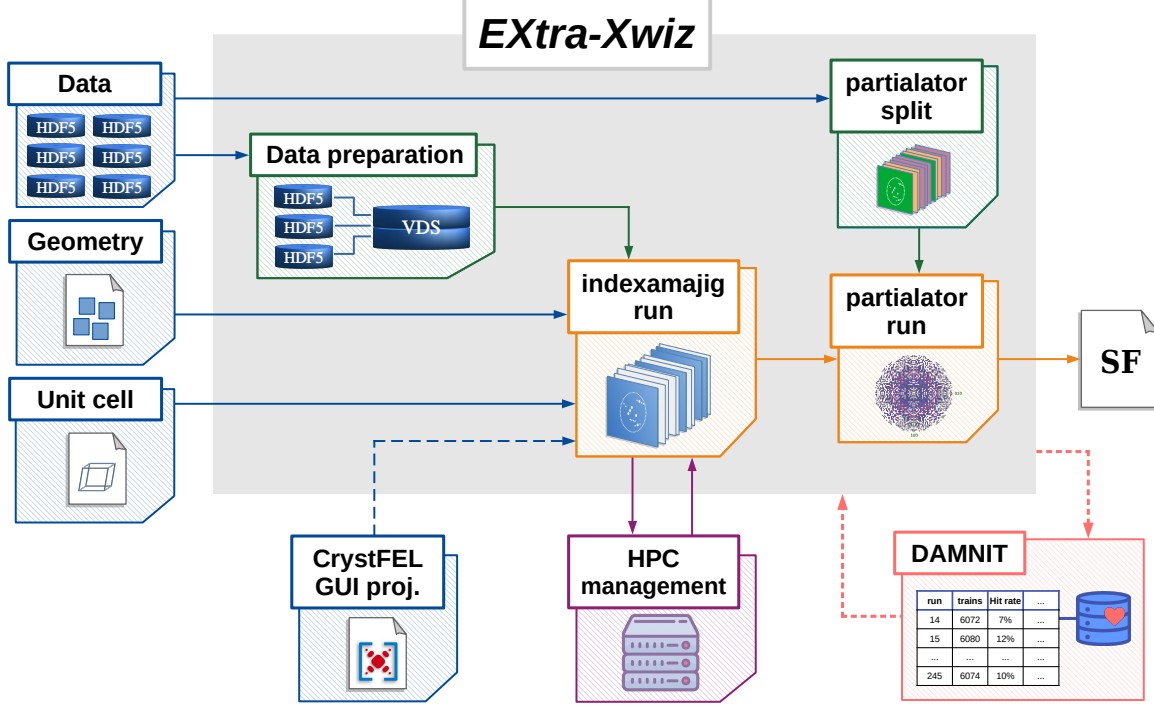

**Figure 1.** Schematic of *EXtra-Xwiz* workflow. The input expected by *EXtra-Xwiz* consists of (i) experimental data ("Data" block), (ii) a detector geometry description ("Geometry" block), and (iii) an optional unit cell file ("Unit cell" block). Data are organized into a single HDF5 file with a "virtual" layout ("Data preparation" block) and processed by *CrystFEL* ("indexamajig run" block). Input parameters for *CrystFEL* can be either specified in the configuration file or imported from the *CrystFEL* graphical user interface project file ("CrystFEL GUI proj." block). The former is executed on the high-performance computing (HPC) cluster Maxwell ("HPC management" block). Afterwards, the output reflections are postprocessed by the *partialator* program from the *CrystFEL* suite ("partialator run" block), which produces a unique set of structure factors ("SF" file). Reflections can be split by *EXtra-Xwiz* into custom subsets ("partialator split" block) using conditions derived from the input data (see text for details). Automatic execution of *EXtra-Xwiz* and harvesting of metadata can be obtained by exploiting the *DAMNIT* tool ("DAMNIT" block). Adapted from [43].

The first step of the *EXtra-Xwiz* pipeline is to provide the data in a format suitable for processing by *CrystFEL*. *CrystFEL* can read diffraction data stored in the Crystallographic Binary Format (CBF) [48] or the Hierarchical Data Format v.5 (HDF5) [49]. For the latter, various standards are supported, including NeXus [50,51] and the native format of the Coherent X-ray Imaging Data Bank (CXIDB) [52].

Due to technical requirements, data from each module of the X-ray detectors at the EuXFEL are saved to separate files. As this process is not compatible with *CrystFEL*, the library *EXtra-data* [53] is used to generate a single HDF5 file with a "virtual" layout (virtual data set file or VDS) containing references to the relevant original EXDF data. This processing step, as illustrated in Figure 2, is represented as a block "Data preparation" in Figure 1 and corresponds to the section "[data]" in the configuration file.

To cope with the variable spatial configurations of detector modules, *CrystFEL* uses a generalized detector representation — the geometry file [29]. This describes the position of each detector pixel in the laboratory coordinate system, as well as the pixel size and the X-ray beam energy. It also contains information on the internal HDF5 paths of the image

data set and any bad pixel masks in the input data file. The geometry file, represented with the block "Geometry" in Figure 1, is specified in the section "[geom]" of the *EXtra-Xwiz* configuration file.

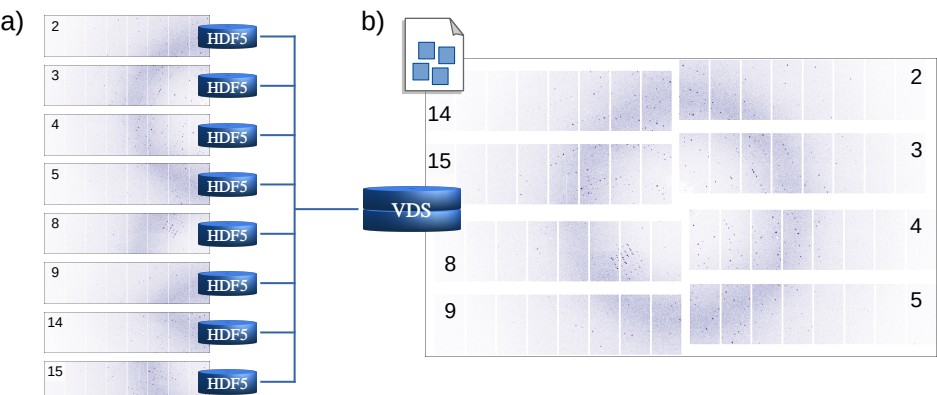

**Figure 2.** Schematic representation of the virtual data set file (VDS) creation, corresponding to the "Data preparation" block in Figure 1. (**a**) Data from each module of EuXFEL X-ray detectors are stored as a different respective file. (**b**) Data from all modules assembled as a single array in a VDS file, corresponding to a single detector frame. A geometry description file is used to spatially arrange modules consistently with the physical detector layout. In this example, only the eight inner modules are displayed.

If prior information on the unit cell is known, it can be used to filter autoindexing results that agree with the expected reference. Such reference unit cell parameters ($a$, $b$, $c$, $\alpha$, $\beta$, $\gamma$) can be specified in the unit cell file and be schematically represented with a block "Unit cell" in Figure 1. It can be provided to *EXtra-Xwiz* in the "[unit_cell]" section of the configuration file. Some of the indexing methods can derive the crystal symmetry even without prior information on the lattice. In this case, *EXtra-Xwiz* can be run without specifying the unit cell file, as is illustrated in Figure 3. After the indexing step, the *cell_explorer* graphical user interface (GUI) will be launched to display the histograms of the resulting lattice parameters. This allows users to determine the cell parameters with a fit and to save them into a unit cell file, which will be expected by *EXtra-Xwiz* to continue processing the data as shown in Figure 1.

The main processing and reduction of the SFX data in *EXtra-Xwiz* is performed with the *indexamajig* tool from *CrystFEL*. It is used for finding the location of Bragg peak candidates in each detector image, indexing the patterns, and integrating peak intensities. For each image frame, *CrystFEL* writes information on the identified Bragg peak candidates, lattice parameters found by the auto-indexing algorithm, and integrates the intensities into a so-called *stream* file in the form of a plain text. This processing step is represented with a "indexamajig run" block in Figure 1. Parameters for *indexamajig* should be specified in the "[indexamajig_run]" section of the *EXtra-Xwiz* configuration file.

Identifying optimal parameters for the Bragg peaks search is crucial for successful indexing. If too many true peaks are missed or too many false peaks (for example, from the detector background or misbehaving detector pixels) are included, the indexing algorithm may fail completely or identify wrong unit-cell parameters and crystal orientations. The sample and data-taking conditions may differ significantly during the experiment, and, therefore, it is a good practice to regularly tune the parameters. The graphical user interface of the *CrystFEL* suite offers a convenient tool for performing such a preparatory step based on visual inspection of a few of the image frames. A session of the *CrystFEL* GUI can be stored into a project file, and *EXtra-Xwiz* provides a command-line tool *xwiz-import-project* for importing parameters from such a file into the *EXtra-Xwiz* configuration file. This procedure is represented by the "CrystFEL GUI proj." block in Figure 1.

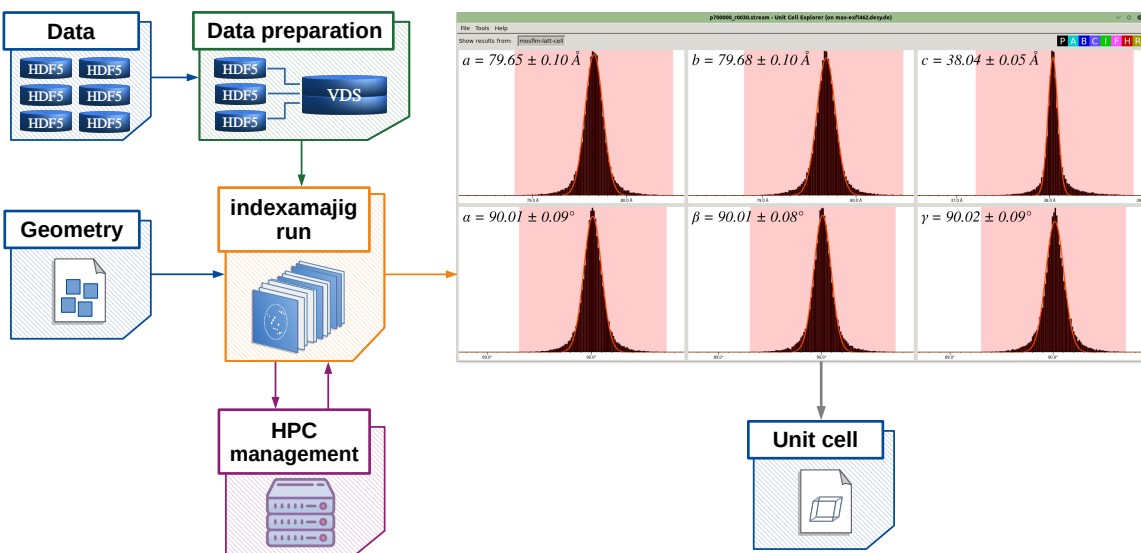

**Figure 3.** Schematic of the *EXtra-Xwiz* workflow used to determine the unit cell in case no initial unit cell file is available. The blocks are the same as defined in Figure 1, except for the *cell_explorer* graphical user interface displayed in the top right corner. After the "indexamajig run", the user is expected to interactively fit the unit-cell parameters through *cell_explorer*. This results in a unit-cell file that can be imported by *EXtra-Xwiz*.

The most time-consuming step of processing SFX data is indexing. Due to the independent nature of the SFX frames, portions of collected data sets can be processed separately, which allows for parallelization. *EXtra-Xwiz* automatically distributes *indexamajig* jobs on the local HPC cluster Maxwell that is operated with the *Slurm* [54] batch-queue system. The results are then concatenated into a single *stream* file. This is represented by the "HPC management" block in Figure 1. In the configuration file, the name of the *Slurm* partition, as well as the number and expected duration of the distributed tasks, have to be specified under the "[slurm]" section.

After indexing, reflection intensities from the whole data set are merged for each symmetrically unique reflection. This task is accomplished by the *CrystFEL partialator* program, which can determine scaling factors for individual measurements, correct for partiality, and postrefine the model parameters for each crystal [55]. This process is usually repeated in an iterative procedure to bring the individual measurements into agreement [30,55]. The *partialator* program operates on the content of the *stream* file and therefore does not require the initial SFX data. This data analysis step corresponds to the "partialator run" block in Figure 1 and the "[merging]" section of the *EXtra-Xwiz* configuration file. Merged reflections are stored in the *hkl* file format as plain text, which can be imported into most structure-solution packages. The output from *partialator* is used by *EXtra-Xwiz* to calculate figures of merit using *check_hkl* and *compare_hkl*. The completeness of the unique reflections and signal-to-noise ratio (SNR) $I/\sigma(I)$ are estimated with *check_hkl* from a single list of reflections that is merged from all of the available data. To calculate the correlation coefficients $CC_{1/2}$ and $CC^*$, as well as the R factor $R_{\mathrm{split}}$, the reflection observations for each unique Miller index are randomly split before being merged by *partialator* into two halves. Both of the halves of data sets are merged separately, and their correlations are computed with *compare_hkl*. The values of the listed FOMs for all data and outer reflection shells are stored by *EXtra-Xwiz* into a table in a summary file.

Certain types of SFX experiments are performed by varying some external parameters, which results in data subsets having different characteristics. These include time-resolved SFX experiments, in which sample states are excited, for example, by visible light (*pump on*) are probed by X-rays as a function of time. The interpretation of such data usually relies on the difference between the subsets. It is therefore crucial to ensure that all of the other

experimental conditions, e.g., the X-ray beam fluence, as well as data processing, are kept as close as possible between the sets. To satisfy this, *pump on* frames are often collected and interleaved with the *pump off* frames within the same train of X-ray pulses. In this case, *partialator* can be used to split reflections after scaling and postrefinement and prior to merging. This option requires a text file with a "data set identifier" (e.g., the pump status) for each frame in the input data. *EXtra-Xwiz* produces such files either using the frame pattern information specified by a user or by automatically utilizing information from a diode, which records the signal from the pump laser. This procedure is represented by the "partialator split" block in Figure 1. The parameters for the splitting have to be specified in an optional "[partialator_split]" section of the *EXtra-Xwiz* configuration file. Merged reflection *hkl* files, as well as FOMs, are produced for each subset, as well as for the overall data set.

The execution of *EXtra-Xwiz* can be automatically triggered as soon as experimental data become available by exploiting its integration with the software *DAMNIT* [56]. Furthermore, *DAMNIT* provides a graphical interface for the *EXtra-Xwiz* results.

## 3. Data Processing with EXtra-Xwiz via Example

This section describes an example of processing SFX data from hen egg-white lysozyme (HEWL) microcrystals collected by the SPB/SFX instrument using the AGIPD detector (run 30, proposal 700000). Only basic knowledge of the Unix commands and environment are expected from the reader. Lines starting with a "$" indicate commands which should be executed in a Unix shell. For readers who are not users of the European XFEL, the Virtual Infrastructure for Scientific Analysis (VISA) [57] service can be used, and additional instructions are provided in Section 3.2.

Access to the Maxwell cluster is exclusive to EuXFEL users, and detailed instructions on how to connect to it can be found in the EuXFEL Data Analysis user documentation [58]. In general, a user with an active account can connect to one of the interactive cluster nodes:

```
$ ssh <user name>@max-exfl-display.desy.de
```

To start using *EXtra-Xwiz*, a dedicated module has to be loaded in the cluster with the following:

```
$ module load exfel EXtra-xwiz/crystals2023
```

As mentioned in Section 2, *EXtra-Xwiz* requires a configuration file in the TOML format [59] for its operation (a detailed description of all the available configuration options is available in the *EXtra-Xwiz* documentation [60]). It should be named "xwiz_conf.toml", and a template of such a file can be generated by starting the pipeline for the first time in an empty folder with the following command:

```
$ xwiz-workflow
```

The configuration file contains parameters for each of the pipeline processing steps that are organized into sections such as "[data]", "[geom]", "[unit_cell]", "[indexamajig_run]", and "[merging]". A copy of the configuration file used in this example, along with all of the other files needed for the pipeline execution, can be downloaded from [47].

The data to be processed by the pipeline can be specified with just a proposal number (the unique identifier of an experiment) and a list of runs in the "[data]" section of the configuration file. In this example, we use open data collected in the context of the experiment published in [61]:

```
[data]
proposal = 700000
runs = [30]
```

It is possible to select a subset of frames from each run with an optional `frames_range` parameter in the same section; see, for example, the following:

```
frames_range = {start = 0, end = 200000, step = 1}
```

This parameter has values that are organized into a dictionary similar to the Python range object, but which is inclusive for the end value, with `end = -1` representing the last frame of the run.

For the purpose of reproducing the SFX analysis, *EXtra-Xwiz* supports a list of different versions of the *CrystFEL* suite, which can be selected in the "[crystfel]" section:

```
[crystfel]
version = '0.10.2'
```

Currently, recent major *CrystFEL* versions are available, as well as a "maxwell_dev" option, which corresponds to the constantly updated installation of the latest *CrystFEL* version.

The geometry and unit cell parameters files should be provided to the pipeline in the "[geom]" and "[unit_cell]" sections, respectively:

```
[geom]
file_path = "agipd_p700000_r0030.geom"
[unit_cell]
file_path = "hewl.cell"
```

A representative detector frame is shown in Figure 2b. The detector modules are positioned in the laboratory frame layout with the use of the *EXtra-geom* library [62].

In the event that the unit cell of the sample is not known prior to the analysis, it can be generated by setting the option `file_path = "none"` as described in Section 2. During the *EXtra-Xwiz* session, after the indexing step, an interactive *cell_explorer* session will start. At this point, the user is expected to determine the cell parameters from the histograms of the indexing results (as explained in [30]), and to then save them into a unit-cell file, which will be requested by *EXtra-Xwiz*. This procedure is illustrated in Figure 3.

The parameters for Bragg peak identification and indexing using the *indexamajig* program have to be specified in the "[indexamajig_run]" configuration block:

```
[indexamajig_run]
resolution = 4.0
peak_method = "peakfinder8"
peak_threshold = 800
peak_snr = 5
index_method = "mosflm"
integration_radii = "2,3,5"
...
min_peaks = 10
extra_options = "--no-non-hits-in-stream"
```

The documentation regarding all of the *indexamajig* options can be found in [46]. In the current state of *EXtra-Xwiz*, not all of these options are covered by the default configuration file parameters, and if any of such options are required for data processing, they can be specified in the string using the `extra_options` parameter. Data processing with *indexamajig* is the most time-consuming step of the whole pipeline, but the computations are usually performed in parallel on multiple nodes of the Maxwell cluster. Cluster partition, the number of nodes to use in parallel and maximum expected duration of the individual jobs, should be specified under the "[slurm]" section of the configuration file:

```
[slurm]
partition = "upex"
n_nodes_all = 20
duration_all = "10:00:00"
```

For testing the pipeline on a small subset of data (e.g., a hundred frames) without exploiting the *Slurm* jobs scheduler, it is advised to select the "local" partition. In this case, *EXtra-Xwiz* will run *indexamajig* on the same node that the pipeline is running.

The reflection intensities obtained from the Bragg peaks indexing are merged and postrefined with the *partialator* tool, and the required parameters have to be specified under the "[merging]" block of the *EXtra-Xwiz* configuration:

```
[merging]
point_group = "422"
scaling_model = "unity"
scaling_iterations = 1
max_adu = 100000
```

Point groups corresponding to the symmetry groups of the crystallized samples can be identified with the table the in *CrystFEL* documentation [63].

In the case of the time-resolved SFX experiments, *pump on* (sample illuminated with the "pump" laser) and *pump off* (sample in the nonexcited state) frames are processed in the same manner and separated only upon reaching the merging step of *partialator*. As already mentioned in Section 2, for such a separation, the *partialator* requires an additional input file that accordingly labels each frame of the input data. *EXtra-Xwiz* can generate such a file according to parameters specified in the "[partialator_split]" block. Let us assume that the machine only delivers at 1/8th of the 4.5 MHz maximum repetition rate, and the detector is configured to record only these pulses. The sample is illuminated by infrared light at every third delivered pulse (i.e., the 24th pulse assuming a 4.5 MHz operation), thereby resulting in the following labels: "pump_on pump_off pump_off pump_on ...". The latter can be set in the configuration as follows:

```
[partialator_split]
execute = true
mode = "by_pulse_id"
[partialator_split.manual_datasets]
pump_on = {start=0, end=-1, step=24}
pump_off = [{start=8, step=24}, {start=16, step=24}]
```

Any user-defined set of labels can be specified with a corresponding list of inclusive range-like dictionary objects or pulse ID values. Usually, in time-resolved experiments, a diode is used to record data relative to the state of the pump laser. *EXtra-Xwiz* can utilize the signal from this diode and automatically generate labels accordingly if the `mode` parameter is set to either "on_off" or "on_off_numbered":

```
[partialator_split]
execute = true
mode = "on_off_numbered"
xray_signal = ["SPB_LAS_SYS/ADC/UTC1-1:channel_0.output", "data.rawData"]
laser_signal = ["SPB_LAS_SYS/ADC/UTC1-1:channel_1.output", "data.rawData"]
```

The difference between the "on_off_numbered" and the "on_off" mode is that, in the former case, consequent events of the same kind (e.g., pump off) are identified by an increasing number. For the example given above, the "on_off_numbered" labels are "on_1 off_1 off_2 on_1, ...". Paths to the diode data specified for `xray_signal` and `laser_signal` are provided by beamline scientists. As the data used in this tutorial does not originate from the pump–probe experiment, the splitting into data sets does not make sense and should be avoided by either setting `execute = false` or simply removing the "[partialator_split]" block from the configuration file.

After all the configuration parameters have been set, the *EXtra-Xwiz* pipeline can be executed in an automatic mode using the following:

```
$ xwiz-workflow -a
```

Without the "-a" ("-automatic") optional argument, the pipeline will verify each configuration parameter with a user in the interactive procedure. When the *EXtra-Xwiz* operation finishes, it will generate a summary file containing information on the processed data statistics and FOMs; see, for example, the following:

```
Step #    d_lim    source        N(crystals)    N(frames)    Indexing rate [%]
1         1.6      indexamajig    46899          639616       7.3
...
Crystallographic FOMs:
overall    outer shell
Completeness                    100.0            100.0
Signal-over-noise               4.224            0.99
CC_1/2                          0.8974           0.03244
CC*                             0.9726           0.2507
R_split                         27.28            80.6
```

These results are only provided to demonstrate the capabilities of the pipeline and could be improved, for example, by tuning the selected parameters and by processing more runs of the collected data. The resulting *CrystFEL stream* file can be found in the same folder, and the output *hkl* file with full FOMs tables can be found in the "partialator" folder.

### 3.1. Automatic Scan over EXtra-Xwiz Configuration Parameters

Sometimes, it is required to run the *EXtra-Xwiz* pipeline by modifying one or multiple configuration parameters over a list of values, which is performed, for example, to assess the sensitivity of a given parameter. For such a use case, a *xwiz-scan-parameters* tool has been developed. Similar to *xwiz-workflow*, it requires a configuration file, and a template is generated on the first use of the tool in an empty folder:

```
$ xwiz-scan-parameters
```

The configuration file "xwiz_scan_conf.toml" for the parameters scan tool consists of four sections: "[settings]", "[xwiz]", "[scan]", and "[output]". The main parameter of the "[settings]" block is `xwiz_config`, which determines the path to the initial *EXtra-Xwiz* configuration file.

The "[scan]" section can contain any number of subsections. Each subsection will be treated as the next level of a nested loop; therefore, the number of iterations in each scan are multiplied. The names of the parameters within each scan subsection represent the full names of the parameters in the *EXtra-Xwiz* configuration files, and their values should contain either a list or an inclusive range-like dictionary of values to scan over. If multiple parameters are listed within one scan, they have to contain the same number of values, which will be modified simultaneously at each step of the scan; see, for example, the following:

```
[scan.SNR]
'indexamajig_run.peak_snr' = {start = 3, end = 7, step = 2}
'indexamajig_run.peak_threshold' = [1000, 800, 700]
```

Here, a scan with three iterations is defined: first an SNR value of three will be used with a threshold of 1000; next, an SNR of five will be set with a threshold of 800; and finally an SNR of seven will be set with a threshold of 700.

In the "[output]" section, the file names to store the tables with the results can be specified. When the configuration file is ready, the parameters scan can be started by running the *xwiz-scan-parameters* in the same folder. It will run *EXtra-Xwiz* sequentially over all of the scan iterations, as well as collect data processing statistics and the overall values of the FOMs for each scan step into a table similar to this one:

```
                                index_rate(%)  ...  cc_half  cc_star  r_split
     peak_snr  peak_threshold
            3            1000          0.080  ...    0.051    0.313   105.40
            5             800          7.332  ...    0.894    0.972    27.44
            7             700          7.219  ...    0.919    0.979    25.81
```

*3.2. Running EXtra-Xwiz Tutorial Using VISA*

VISA is a platform for cloud-based data analysis, which was developed by the Institute Laue-Languevin and deployed at several European photon and neutron facilities [57]. Its documentation can be found in [64].

In order to perform this tutorial using VISA, an instance containing an *EXtra-Xwiz* installation can be generated and used in a web browser:

1.  Navigate to https://visa.xfel.eu, accessed on 16 October 2023 [65];
2.  Click "Create a new instance";
3.  Click "Search for experiments", and select the proposal "p700000-SFX on Hen egg-white lysozyme, AGIPD detector";
4.  Click on the "EXtra-Xwiz_Crystals2023" environment;
5.  Choose the virtual hardware;
6.  Create the instance.

There is no need to load any additional modules; the tutorial described above applies, except for the distribution of computations on the HPC cluster, which is not accessible using VISA. Because of that, the following should be used:

```
[slurm]
partition = "local"
[indexamajig_run]
n_cores = 1
...
```

Be careful, as this would result in a very slow processing of the input data. Therefore, instead of running *EXtra-Xwiz* on the entire run, a preselection of "good frames" can be used by specifying the following in the "[data]" section:

```
[data]
...
frames_list_file = "indexed_p700000_r0030.lst"
```

Please note that the "indexed_p700000_r0030.lst" file has been produced specifically for this VISA example, and this option is never used in the actual processing of the experimental data.

## 4. Discussion and Outlook

Photon sources based on XFELs, and especially the high-repetition rate ones such as in the European XFEL, offer incredible potential for serial femtosecond crystallography. This comes at the cost of complicated setups, which are also reflected in the data structures, as well as overwhelming data rates that need to be tackled by nonroutine data processing. A challenge that requires expert knowledge and/or iterative processing steps lies in the optimization of the parameters, such as a minimum signal-to-noise ratio, peak finder thresholds, et cetera. Technically, as these parameters are passed to the *CrystFEL* programs through a configuration file, iterative runs require a re-editing of the configuration, which can quickly become tedious. Currently, *EXtra-Xwiz* offers two ways to simplify this process: first, the software includes a grid search, as described in Section 3.1, that can scan over some parameter space. This requires some estimation of the reasonable parameter ranges, as well as enough time to compute the necessary grid nodes. Second, peak-finding parameter optimization can be performed manually, and with visual feedback, using the *CrystFEL* GUI. The results of a GUI session can be stored to a project file to be used by *EXtra-Xwiz* for the

batch processing of one or more entire runs. Both of these approaches have their limitations, particularly those related to the interpretation of their results by inexperienced users. In contrast to brute-force searches, optimization methods, for example, those based on artificial intelligence, can be employed. In particular, we developed a method based on Bayesian optimization that optimizes the *EXtra-Xwiz* parameters by maximizing the indexing rate, as well as reduces the need for expertise in the interpretation of the results [66]. This solution is being deployed at the EuXFEL and integrated into *EXtra-Xwiz*. The same approach can also be used to tune the detector geometry representation in laboratory space. Indeed, the overall SFX workflow can greatly benefit from automating this task. At the moment, this is largely done manually. The operator is typically using graphical software for the visual centering and quadrant fitting of the powder diffraction rings, followed by iterations of *indexamajig* and *geoptimizer* [67], for maximizing the yield of frames that can be indexed.

As a more long-term outlook, we strive to give users an end-to-end experience, that is, a pipeline covering all of the steps from detector images to an atomic structure built into the reconstructed electron density. For this purpose, downstream modules handling the program execution related to the tasks of crystallographic phasing, model building, and (preliminary but automated) structure refinement need to be included to the *EXtra-Xwiz* framework. The integration of different SFX processing software, such as DIALS, as alternatives to *CrystFEL* would give users more options and allow a more objective, respectively comprehensive, data analysis. Such integration is planned, as it will be facilitated by the modular nature of *EXtra-Xwiz*. It should be pointed out that challenging problems, particularly with respect to de novo structures, will require user experience for decisions along the way, for instance, these include which phasing method to apply or which search model to use for molecular replacement. This means that the design of the workflows has to be well-thought, thereby allowing user intervention where required and a good balance of parameters with reasonable defaults versus expert input, thereby boiling the process down to a useful semiautomatic approach. For this to be achieved, an improved user interface is being designed.

## 5. Conclusions

*EXtra-Xwiz* has been designed to support scientists performing SFX at the European XFEL. It handles tasks such as data preparation, the discrimination of data based on the pump status, and parallel processing, which often pose a significant barrier to inexperienced users.

**Author Contributions:** Conceptualization, F.D., H.F. and L.G.; methodology, O.T., F.D., H.F., D.E.F.d.L., M.S., E.S., J.J.W. and L.G.; software, O.T. and F.D.; validation and data curation, J.E., S.K., H.J.K., F.H.M.K., D.V.M.M., A.R., E.S. and R.d.W.; writing—original draft preparation, O.T., F.D. and L.G.; writing—review and editing, R.J.B., J.E., H.F., D.E.F.d.L., F.H.M.K., A.P.M., M.S., R.d.W. and J.J.W.; visualization, O.T., F.D. and L.G.; supervision, F.D., H.F. and L.G.; project administration, L.G. All authors have read and agreed to the published version of the manuscript.

**Funding:** This publication was partially supported by the consortium DAPHNE4NFDI in association with the German National Research Data Infrastructure (NFDI) e.V. NFDI is financed by the Federal Republic of Germany and the 16 federal states and the consortium is funded by the Deutsche Forschungsgemeinschaft (DFG, German Research Foundation)—project number 460248799. The authors would like to thank for the funding and support. Furthermore, thanks go to all institutions and actors who are committed to the association and its goals.

**Data Availability Statement:** Data sets used for the tutorial are freely accessible through the European XFEL VISA instance [65] and, for European XFEL users, at [68].

**Acknowledgments:** We appreciate the contributions to the construction, maintenance, and operation of the European XFEL and scientific instruments by the many people who are not listed as authors. We would like to thank all of the members of the EuXFEL Data Analysis group for their continuous development and improvement of software tools and libraries for efficient data analysis. We are grateful to Philipp Schmidt for valuable feedback on the manuscript and his support. We would like

to especially thank Thomas White, the main developer of the *CrystFEL* suite, for the great software for the analysis of serial crystallography data and his invaluable input and feedback to our project. The authors would like to acknowledge Oleksandr Yefanov, whose constructive criticism allowed us to shape a clear vision for the evolution of our framework. We would also like to express gratitude to the users of our software, who provided continuous feedback and useful suggestions.

**Conflicts of Interest:** The authors declare no conflict of interest.

## Abbreviations

The following abbreviations are used in this manuscript:

| | |
|---|---|
| SFX | Serial Femtosecond Crystallography |
| XFEL | X-Ray-Free Electron Laser |
| EuXFEL | European XFEL facility |
| EXDF | EuXFEL Data Format |
| AGIPD | Adaptive-Gain Integrating Pixel Detector |
| LPD | Large Pixel Detector |
| HPC | High-Performance Computing |
| HDF5 | Hierarchical Data Format v.5 |
| CBF | Crystallographic Binary Format |
| CXIDB | Coherent X-Ray Imaging Data Bank |
| GUI | Graphical User Interface |
| VDS | Virtual Data Set File |
| FOM | Figure Of Merit |
| SNR | Signal-To-Noise Ratio |
| HEWL | Hen Egg-White Lysozyme |
| VISA | Virtual Infrastructure for Scientific Analysis |

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
