# Peer review of "EXtra-Xwiz: A Tool to Streamline Serial Femtosecond Crystallography Workflows at European XFEL"

_crystals, doi:10.3390/cryst13111533_

Round 1
Reviewer 1 Report
The manuscript from Turkot et al., on streamlining CrystFEL processing is very relevant in the context of making serial crystallography routine with non-expert users. I highly recommend to publish this manuscript.
Comments:
1. An estimation on processing time required for EXtra-xwiz software for a given XFEL dataset is needed. This will allow readers to understand what semi-automation refers to.
2. Currently the software is applied only to harness CrystFEL. What about cctbx.xfel or DIALS? Having multiple softwares available for automated processing will allow users to obtain an un-biased complete picture of the dataset.
Author Response
Dear Reviewer,
Thank you a lot for providing a very positive and valuable review.
We would like to reply to your comments:
> 1. An estimation on processing time required for EXtra-xwiz software for a
> given XFEL dataset is needed. This will allow readers to understand what
> semi-automation refers to.
Although we agree that such information would be beneficial for readers, we are
afraid it is not possible to provide an exact timescale. The main reason is
that the time required to process a set of data of the same size vary
considerably, up to perhaps an order of magnitude, depending on the chosen
indexing algorithm, hit fraction of the sample or selected data processing
parameters. Given that the workflow manager has a negligible overtime, the
total runtime is almost exclusively defined by CrystFEL, especially the
indexing step. Therefore, we would prefer not to report on any number.
> 2. Currently the software is applied only to harness CrystFEL. What about
> cctbx.xfel or DIALS? Having multiple softwares available for automated
> processing will allow users to obtain an un-biased complete picture of the
> dataset.
Thank you for this valuable comment, we indeed have plans to include DIALS in
the future releases of EXtra-Xwiz, therefore we've decided to expand what was
already written in section 4 as:
"The integration of different SFX processing software, such as DIALS, as
alternative to CrystFEL would give users more options and allow a more
objective, respectively comprehensive, data analysis. Such integration is
planned, as it will be facilitated by the modular nature of EXtra-Xwiz."
which are new lines 392-395 in the updated manuscript.
Please take a look at the attached updated version of the manuscript with all
changes highlighted for convenience.

Reviewer 2 Report
Review notes:
1) Hierarchical Data Format v.5 (HDF5) is described in line 132 but first mentioned in line 89
2) In figure 1, this reviewer cannot see where EXtra-Xwiz plays its role. It should be obvious to readers from the figure without referring to the legend.
3) Figures are referred both as ‘Figure 1’ and ‘Fig. 1’ - consider being consistent in this way (Line 139 has both uses, ‘Fig 2’ and ‘Figure 1’)
4) In this reviewer’s opinion, the purpose of Figure 2(a) is convoluted by the extra files and would be easier to understand if simplified. If Figure 2(b) is intended to represent a representative detector frame, it is very dark. Also, Figure 2, in my opinion, needs a more descriptive legend.
5) It is this reviewer’s opinion that Figures 2 and 3, which should provide a visual demonstration on VDS organization and unit cell selection by EXtra-Xwiz, do not support a reader’s understanding on the topic. They could be improved to better convey their function or the process by which they work.
6) A description of the proposal on Line 242 would be helpful for new users.
7) It may be worth reiterating what challenge we are referring to in the Discussion opening sentence for readability.
Author Response
Dear Reviewer,
Thank you a lot for providing valuable and detailed comments.
Following them in order:
> 1) Hierarchical Data Format v.5 (HDF5) is described in line 132 but first
> mentioned in line 89
Since the detail that data is stored in the HDF5 format is not significant in
line 89 we've decided to skip it there and introduce it in line 131.
This change is reflected in lines 88-89 of the updated manuscript.
> 2) In figure 1, this reviewer cannot see where EXtra-Xwiz plays its role. It
> should be obvious to readers from the figure without referring to the legend.
Addressed with the update to Figure 1.
> 3) Figures are referred both as ‘Figure 1’ and ‘Fig. 1’ - consider being
> consistent in this way (Line 139 has both uses, ‘Fig 2’ and ‘Figure 1’)
All occurrences of ‘Figure‘ have been replaced with ‘Fig.‘.
> 4) In this reviewer’s opinion, the purpose of Figure 2(a) is convoluted by
> the extra files and would be easier to understand if simplified. If
> Figure 2(b) is intended to represent a representative detector frame, it is
> very dark. Also, Figure 2, in my opinion, needs a more descriptive legend.
Addressed by modifying Figure 2 and it's caption.
> 5) It is this reviewer’s opinion that Figures 2 and 3, which should provide a
> visual demonstration on VDS organization and unit cell selection by
> EXtra-Xwiz, do not support a reader’s understanding on the topic. They could
> be improved to better convey their function or the process by which they
> work.
We have modified Figures 2 and 3 and enriched the figures captions.
> 6) A description of the proposal on Line 242 would be helpful for new users.
Thank you for the comment, we have specified the meaning of the "proposal" term
in lines 240-241 and included the reference to the experiment which resulted in
these data in lines 242-243. We apologize for forgetting this citation in the
previous version of the manuscript.
> 7) It may be worth reiterating what challenge we are referring to in the
> Discussion opening sentence for readability.
Thank you, we have addressed this by including a better introduction to the
Discussion section in lines 361-364 of the updated manuscript.
Please take a look at the attached updated version of the manuscript with all
changes highlighted for convenience.

Reviewer 3 Report
Thia is a very interesting presentation of an integrated and sophisticated strategy and of the corresponding system to implement in a systematic and powerful way what is, in my opinion, the most important use of x-FEL. The strategy is not only effective but very complete, dealing with general technical issues but also with essential details. The presentation will be of interest to the broad research community that woukd like to use these novel techniques. The authors also present interesting news for specific problems: I can mention in particular the Bayesian strategy. In the absence of serious issues, I can recommend publication in the present form
Author Response
Dear Reviewer,
Thank you a lot for providing such a positive review and motivating feedback.
Sincerely,
Authors.